# 1°C warming increases spatial competition frequency and complexity in Antarctic marine macrofauna

David K. A. Barnes [1✉], Gail V. Ashton [2], Simon A. Morley [1] & Lloyd S. Peck[1]

Environmental conditions of the Southern Ocean around Antarctica have varied little for >5 million years but are now changing. Here, we investigated how warming affects competition for space. Little considered in the polar regions, this is a critical component of biodiversity response. Change in competition in response to environment forcing might be detectable earlier than individual species presence/absence or performance measures (e.g. growth). Examination of fauna on artificial substrata in Antarctica's shallows at ambient or warmed temperature found that, mid-century predicted 1°C warming (throughout the year or just summer-only), increased the probability of individuals encountering spatial competition, as well as density and complexity of such interactions. 2°C, late century predicted warming, increased variance in the probability and density of competition, but overall, competition did not significantly differ from ambient (control) levels. In summary only 1°C warming increased probability, density and complexity of spatial competition, which seems to be summer-only driven.

[1] British Antarctic Survey, NERC, Cambridge, UK. [2] Smithsonian Environmental Research Center, Tiburon, CA, USA. ✉email: dkab@bas.ac.uk

For millions of years the Southern Ocean has been one of the most thermally constant of Earth's environments, but is now undergoing multiple, complex, interacting physical changes[1]. This region includes a major centre of considerable, recent warming in the shallows, and this is forecast to be sustained[1]. It will likely drive varied and considerable biological change, which remains little investigated in situ. Most existing knowledge is for responses of individual species, in isolation[2], but cumulative responses at assemblage and community levels, though poorly studied, will likely have greater consequences[3]. There is now a wide literature on indirect impacts of warming on biota (e.g., snow and ice retreat, freshening, and sedimentation from glaciers, among others[4–7]) but few field studies on specifically direct thermal effects. To date, warming impacts have been predicted to change species success[4,8] and the first polar assemblage level data demonstrated increased growth[9]. However, this only occurred in a few species at moderately increased temperature. If sessile animals become larger (owing to increased growth) this is more likely to make space a limiting resource and increase the incidence, and importance, of spatial competition.

In the current study, we investigated how in situ warming impacts physical 'fighting' for space (so called contest competitive interactions), between species in assemblages. This is where the boundaries of colonies/individuals meet others, which leads to either a cessation and redirection of growth by both competitors (a tie or draw) or overgrowth of one (a loser) by the other (winner). To our knowledge, the impact of climate-forcing on spatial competition has not been considered in polar seas. Yet, for species unchanging in growth performance (and even some of those which do increase growth) competitive encounter frequency might be easier to detect and therefore be an earlier measure of response to environmental change. This is because snapshots of the extent of spatial competition can be obtained using still photographs either by SCUBA or Remotely Operated Vehicles. In comparison, growth has to be monitored over long periods of time and compared within species across years. Bryozoans and other encrusting cryptofauna have proved strong model taxa for investigating spatial competition and artificial substrata, in the form of settlement panels, are good experimental surfaces to investigate such encounter dynamics[7,9–13]. To investigate responses to global physical change, the next step is to be able to manipulate one aspect of artificial substrata in situ whilst not altering any others.

Heat controllable settlement panels[9] allow exploration of predicted mid or end-century shallow sea temperature levels in situ, which is enhanced by including several warming regimes (year-round and summer only) and levels (0, +1, and +2 °C). Different levels of warming treatments aid prediction of future responses, but are also useful because warming is geographically highly variable, even around the West Antarctic Peninsula (WAP). Using this apparatus, Ashton et al.[9] found that growth (and per cent cover change) responses varied considerably between warming levels in the six most common recruit species[9]. In particular, a 1 °C temperature rise led to one bryozoan species, *Fenestrulina rugula*, monopolising most space (~60%), despite being a weak spatial competitor (it is out competed and overgrown in physical encounters with most other species it meets)[7]. What does this mean for assemblage dynamics and intra- and interspecific competition for space? Other factors being equal, more-occupied space should increase the incidence and importance of spatial competition. Thus Ashton et al.'s[9] findings led us to hypothesise that (1) competitive encounters per unit area, and the probability of a given individual, or colony, being involved in spatial competition would increase with moderate (1 °C) warming, but less so, if at all, with 2 °C warming. The reasoning behind increased competition with 1 °C but not 2 °C warming was that

Ashton et al.[9] found increased growth with 1 °C but not 2 °C warming—making it more likely that the boundaries of species should come into contact. Our hypothesis (2) was that the spatial dominance of *F. rugula* would lead to more competition involving this species and fewer interactions involving other species (less complexity). Typically, investigation of the impacts of treatments such as warming, compares changes in species composition across treatments[14]. We, however, compared competitive pairings between species (across treatments). We predicted that the similarity of competitor pairings would provide a stronger response signal to warming than mere species composition, as the number of potential competitive interactions between species is the factorial of presence/absence.

We found that panels that were warmed to 1 °C above ambient (either throughout the year or just summer only) increased the probability of spatial competition among encrusting nearshore Antarctic fauna. This level of warming also increased the density and complexity of spatial competitive interactions. In contrast, warming to 2 °C above ambient increased variance (rather than mean) in the probability and density of competition, but competition did not significantly differ from ambient (control) levels. Thus biological responses, in terms of spatial competition, to warming change alter with both level and (seasonal) timing of warming. We found evidence that changes in competitive structure may be detected before changes in species composition, thus panels may be a powerful tool for monitoring early community responses to stressors such as climate change.

## Results

The 16 panels (4 of each treatment type) deployed on the WAP (Fig. 1) were colonised by 5360 encrusting cryptofauna (colony density), which had 4532 spatial competitive encounters. We found that the probability of colonists being involved in spatial competition varied from 0.49 to 0.98 and competition

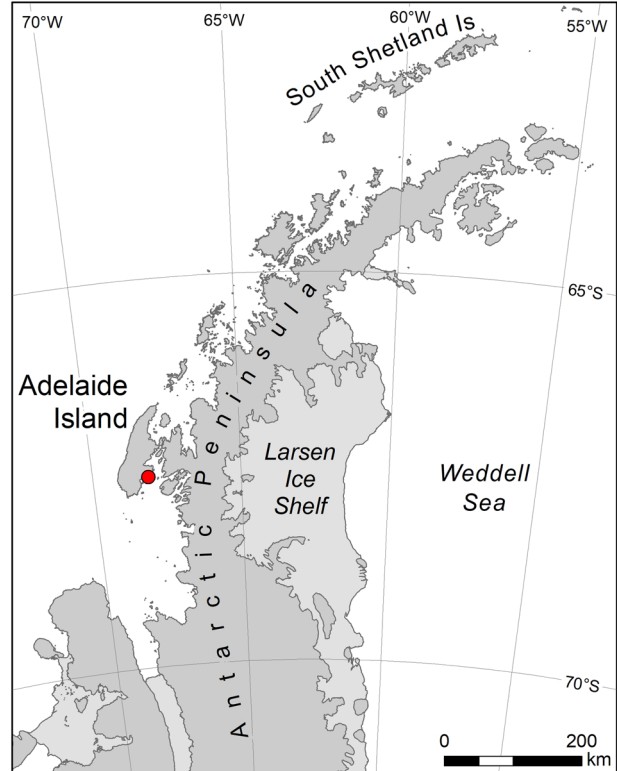

**Fig. 1 Location of deployment site in Antarctica.** Study area. Red dot within this is study site adjacent to Rothera Research station.

density varied from 0.6 to 5.1 competitive encounters per cm$^2$. As hypothesized, compared with controls, both of these measures significantly increased (both Welch's one-way and Games–Howell post hoc comparisons $p < 0.1$) at $+1\,°C$ treatment, but not at $+2\,°C$ (Fig. 2A–D). Panels warmed to $1\,°C$ above ambient (which is the temperature predicted by mid-century)[1] showed that this level of warming significantly increases the probability of cryptofauna encountering contest, spatial competition and that density of such competitive interactions also increased (Fig. 2). Contrary to expectations, spatial competition increased in complexity with warming compared to controls, but only significantly so at $+1\,°C$ (Fig. 2E; analysis of variance (ANOVA) $p = 0.006$, Tukey HSD post hoc comparison $p < 0.1$). Increased complexity of competition means there were more different unique competitor pairings. The 5360 recruits on the panels included 9–16 species on each panel, but variability in species richness and presence of rare species (Fig. 3C & D) was too high within treatments to detect differences between treatments. Significantly more competition was intraspecific at $+1\,°C$ (Welch's one-way $p = 0.02$, Games–Howell post hoc comparisons $p < 0.01$), although the probability of *F. rugula* encountering competition did not change (see Fig. 3E, F respectively).

Overall $+2\,°C$ warming increased the variance of competition, but competition did not significantly differ from ambient (control) levels in terms of probability, density, complexity or any of our other measures. Of note, though was that summer-only warming was not detectably different from year-round warming. Thus, warming for just part of the year is enough to have a measurable and considerable impact on spatial competition amongst shallow cryptofauna. Also of note was that cryptofauna on unheated panels from the same site but in previous years were not significantly different from our controls on any measure of competition, richness or composition. Surprisingly though, the share of competitive encounters involving the most numerous competitor, the bryozoan *F. rugula*, did not significantly change (Fig. 3F), though more encounters were intraspecific at $1\,°C$. Thus there was a higher degree of intraspecific competition with warming but the species involved did not change. Similarity analysis identified a significant response of competition to warming (Fig. 3B).

There was also a response in simple species abundance (community composition) but this was not significant (Fig. 3A; ANOSIM $R = -0.041$, $p = 0.632$). Ordination (nMDS) of competitor interactions across treatments shows how distinct warming treatments were from ambient assemblages (Fig. 3B; ANOSIM $R = 0.4$, $p = 0.002$). Species involved in competitive interactions on panels warmed to $+1\,°C$ either throughout the experiment or during summer only were significantly different from those on unheated panels deployed at the same time (control) or in previous years (previous) (one-way ANOSIM pairwise tests, all $p$ values $= 0.029$). Intraspecific competition between adjacent *F. rugula* colonies explained >70% of significant dissimilarities between groups (SIMPER). Species involved in competitive interactions on panels warmed to $+2\,°C$ were not significantly different from any other treatment (all $p$ values > 0.1). In contrast, species composition data (Fig. 3A) varied much less across treatments than suites of assemblage interactions. Interactions might, therefore, be a more sensitive ecological method to monitor near future and subtle impacts on biodiversity.

## Discussion

Understanding biological responses to physical change around Antarctica is proving difficult; the physical changes are complex in time and space, non-linear and non-conforming to model projections[6]. Warming is just one physical change, though an important one, that has already shown considerable physical and biological impacts on the Antarctic environment. Artificial substrata with controllable heating have proved an important step forward in understanding individual and assemblage responses in terms of their own performance[9,13] but the current study takes this a further step forwards, by exploring interactions between species under climate change scenarios. Our key findings underlying hypothesis 1 were that only $1\,°C$ warming increased the probability of any individual being involved in spatial competition, but this seemed entirely driven by summer only warming. This is important because warming varies considerably across seasons[15]. Across hypotheses 1 and 2, competition density and complexity both increased with warming but only significantly at $+1\,°C$. Confidence with our method and robustness of our single-year data are supported by the similarity across all our previous years' treatments of unheated panels from the same site and our controls (white vs black box symbols in Fig. 2 and 3). Our study thus strongly suggests that, given increased growth and spatial coverage[9], an increase in competition intensity with moderate warming seems likely. Yet intensities of temperate (or tropical) competition, in similar assemblages, are no higher[13]. If organisms encrusting polar coastal substrata were only influenced by warming we would expect increased interspecific and reduced intraspecific competition, moving assemblages towards a competitive regime more typically described from temperate latitudes[13]. However, concurrent raised stresses of sedimentation induced by glacier retreat[5] and ice scour increases induced by reductions in sea ice[7] seem likely to increase disturbance rate of assemblages, potentially counteracting any benefit of increased growth and development promoted by elevated temperature[9].

Assemblages that we (and other researchers[10–13,16]) report from artificial substrates tend to be young, so increases in competition probability and density are potentially only a short-term effect. There is a wider question of how relevant short-term changes in young fouling communities of the shallows are to long-term changes in natural communities. Undisturbed coasts are undoubtedly becoming rare and even around Antarctica icebergs frequently scour the seabed shallower than 100 m depth. This means that pioneer fouling species are becoming more common, abundant and 'the norm' in the shallows around the world because coasts are hotspots of direct and indirect human impact. However, as elsewhere, understanding separate elements of environmental impact even within a specific factor (such as warming) and time scale (such as months-a year) is difficult. The increased complexity reported here contrasts markedly to decreased complexity observed on natural substrata in response to marine warming correlate; increased ice scouring[7]. Polar ice scouring rates are influenced strongly by fast ice duration (less fast ice allows icebergs to travel more, increasing scouring potential), which is in turn influenced by warming[7]. However, no scour marks were seen on panels, those hit by icebergs were completely crushed.

Contrary to the previously described hypothesis, faster growth, driven by moderate temperature increase, may help assemblages recover more quickly, and this might offset the negative impact of more scouring. If new non-indigenous species establish and spread, as reported recently[17], spatial competition intensity and complexity could increase further.

We tried an alternative and widely used way of evaluating and comparing our spatial competition data using ordination (nMDS). With this approach, it was clear that competitor interactions across treatments showed the effects of warming treatments (Fig. 2B) much more distinctly than species composition data (Fig. 2A). This suggests, as hypothesized, that interactions could prove a superior ecological method to monitor biodiversity

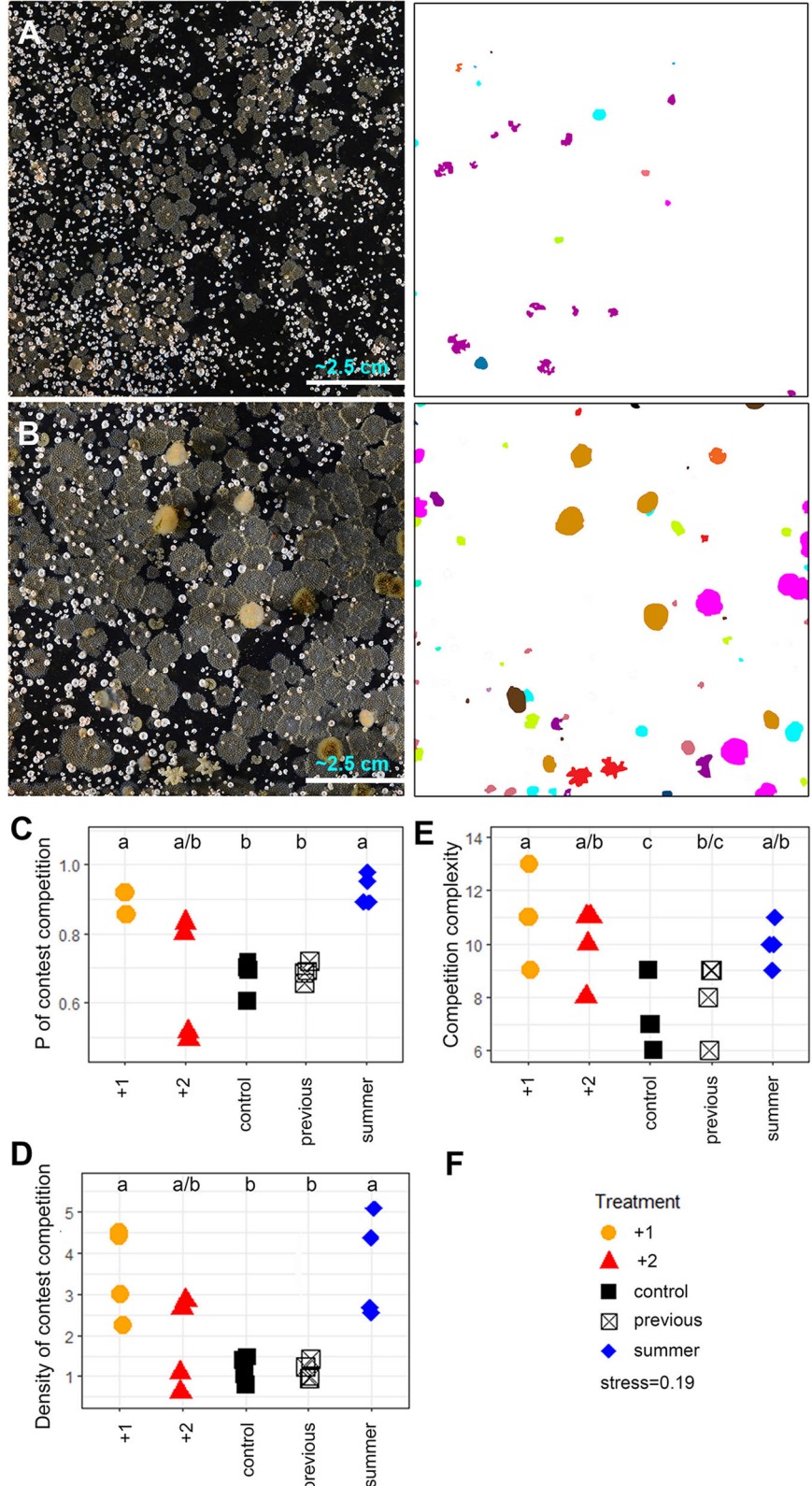

**Fig. 2 Spatial competition on artificial substrata at 15 m depth in Antarctica.** Colonists of panels at ambient (**A**) and +1 °C (**B**) treatments (9.8 ×9.8 cm), together with image analysis frames used to highlight less abundant species (right images). Rarer species are shown in different colours. Probability of individuals encountering competition (**C**), competition density (**D**) and complexity (**E**). Letters above plots **C**–**E** show significant differences between treatments. Degrees of freedom (DF) = 4, $n = 20$ for **C**–**E**; (**C**) Welch's one-way $F = 28.326$ $p = 0.0001$, Games–Howell post hoc tests $p < 0.01$; (**D**) Welch's one-way F = 6.5527 $p = 0.017$, Games–Howell post hoc tests $p < 0.1$; (**E**) ANOVA F = 5.467 $p = 0.006$, Tukey HSD $p < 0.1$. Key to treatment symbols (**F**), +1: warmed to +1 °C; +2: warmed to +2 °C; controls: no warming; previous: deployed previously without warming; summer: warmer to +1 °C in summer (Sep–Mar) only.

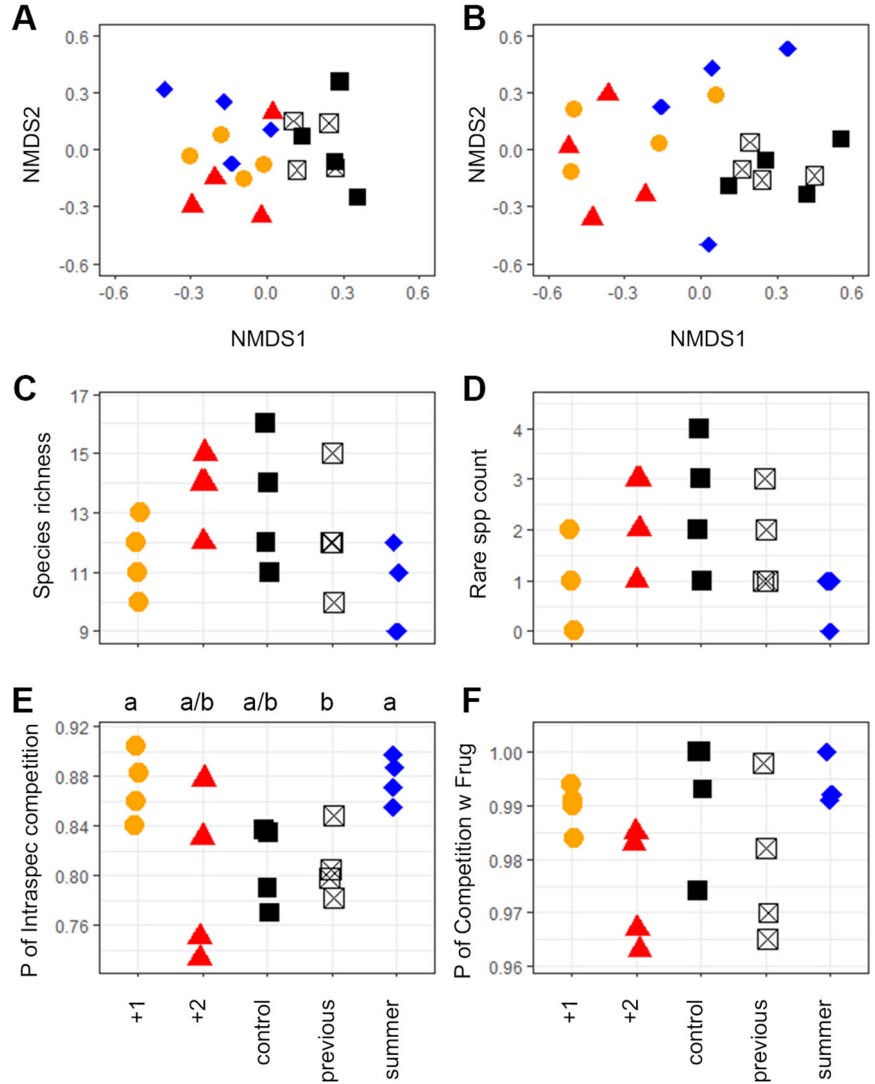

**Fig. 3 Composition, richness and competition in Antarctic encrusting assemblages with differing warming treatments.** Similarity of assemblage composition (**A**, ANOSIM $R = -0.041$, $p = 0.632$, 1000 permutations) and similarity of species involved in competitive encounters by treatment (**B**; ANOSIM $R = 0.4$, $p = 0.002$, 1000 permutations) presented using non-metric multidimensional scaling (key to treatment symbols Fig. 2F). Species richness (**C**) and counts of colonies of rare species (**D**). Probability of individuals encountering intraspecific competition (**E**) and probability of species encountering competition with spatially dominant *F. rugula* (**F**). Letters above plot E indicate significant differences between treatments. Degrees of freedom (DF) = 4, $n = 20$ for **C**–**F**; (**C**) ANOVA $F = 2.659$ $p = 0.073$; (**D**) ANOVA $F = 2.632$, $p = 0.076$; (**E**) Welch's one-way $F = 5.6086$ $p = 0.022$, Games–Howell post hoc tests $p < 0.1$; (**F**) Welch's one-way $F = 2.6113$, $p = 0.13$. Treatments did not significantly influence species richness (**C**), rare species (**D**), or probability of competition with the key species *Fenestrulina rugula* (**F**). Key to treatment symbols (Fig. 2F).

responses to physical change, at least for cryptofauna. It is not easy to assess how relevant our findings of change in competition by sessile species growing on hard surfaces are to sessile species found on soft substrata or mobile species in either habitat. Intuitively, it seems likely that competition for resources within soft-sediment assemblages might similarly increase with moderate warming, and that increased growth and space coverage by sessile fauna should provide more food for mobile species. Measuring contest competition on panels is relatively easy, as panels can be removed, preserving a snapshot of all competitive interactions for the duration of lengthy analyses. Determining how competition might change in other types of biota would be much less easy in Antarctica. Variations in species composition may be observed at later stages of community development, that were not captured here. Even in low-diversity situations, species are unlikely to respond in isolation to physical forcing, so considerable progress is required to understand interactions that

drive assemblage, community or even ecosystem responses. This, of course, is more technically demanding and time-consuming than assessment of species in isolation[18], especially in remote locations like Antarctica. However, striking in situ temperature responses of biodiversity in growth[9] and competition performance show the importance of considering assemblages and their functionality, not just their presence or range[6].

## Methods
**Study site and apparatus**. Our experimental apparatus (Fig. 2A) was established at a WAP site (Fig. 1), near Rothera Research Station, in the region of fastest Southern Ocean warming detected to date[9,10]. Our experimental apparatus and field protocol is described in detail by Ashton et al.[9]. Artificial substrata, in the form of settlement panels, were constructed with metal heat traces embedded in polyvinyl chloride (PVC) to allow control of the temperature on the panel surface (micro abraded high-density PVC). Using SCUBA, each panel was placed on boulder and cobbled seabed at 15 m depth, connected by 100 m of cable to a shore-based control unit. When in situ in Antarctica, panels were controlled using resistors on a shore-based AC supply.

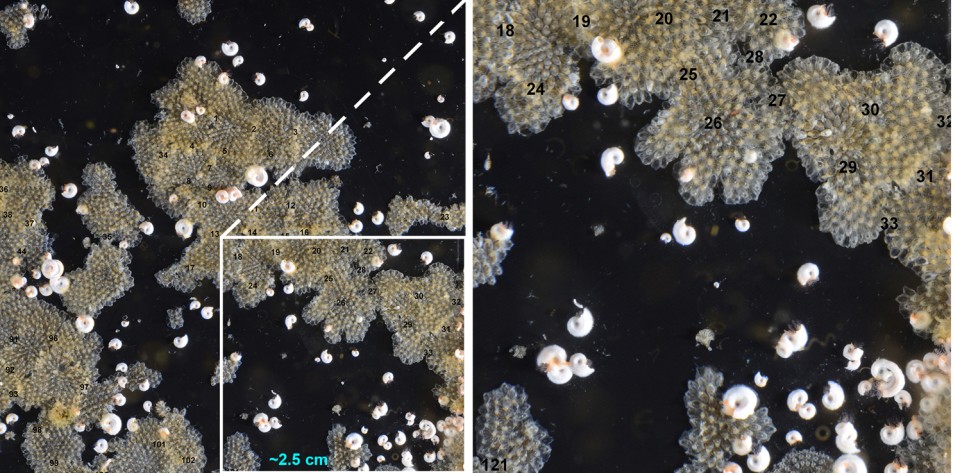

**Fig. 4 Example panel area showing spatial competition by bryozoans.** Competitive encounters were each marked with a number as they were counted to ensure none were missed or counted twice. The left image is zoomed into one quarter of a panel (image size 4.9 × 4.9 cm) and the right one further zoomed into one-sixteenth of a panel.

**Apparatus performance trials**. Aquarium trials prior to field deployment, showed that 14.2 V and 20.1 V of input (verified using an inline voltmeter) produced consistent 1 °C and 2 °C warming, respectively[9]. The temperature of the panels and the water immediately above them varied over the year with ambient environment variability but the warmed panels were always 1 °C and 2 °C warmer than ambient temperatures. We measured the temperature change immediately above the panel surfaces, rather than the PVC surface itself. Trials of the evenness of the warming of the boundary layer across the 9.8 × 9.8 cm central area of the panels, where colonisation was monitored, were performed in a flow flume during the design phase, during in situ shallow trials and in aquaria post deployment. Digital thermometers were set up in the aquaria so that the distance from the panel front could be dialled in by 1 mm at a time. Panels were heated evenly as 50 observations across a panel set at 1 °C of warming showed the layer of water 2 mm above the surface of the panel was heated by 1.01 ± 0.03 °C (mean ± SE). No animals in the experiment (study period of 2014–2015) grew beyond this 2 mm layer. In the flume trials the shape of the warmed boundary layer was slightly cooler downstream, which would not have been the case if there had been a considerable convective lift.

**Seawater temperatures**. Seawater temperatures at the study site ranged from −1.7 to −0.2 °C at 15 m depth over the study period of June 2014 to March 2015. There was a higher variability in temperature in summer with a monthly maximum variation of ~1.7 °C, which reduced to a 0.2 °C variation within winter months. Thus our control panels were at these ambient temperatures throughout and our treatments were maintained 1 °C and 2 °C above these temperatures throughout. If the presence of convective currents rising up from panels had been detected it could mean that panel colonists had less access to food than those in ambient conditions because such currents could disrupt feeding currents generated by organisms. Water movement in the shallows around the study site is >20 cm sec$^{-1}$ and would dwarf any potential convective lift (given that even aquarium trials failed to detect this) and thus minimising any potential food (nano phytoplankton, e.g., ciliates and flagellates) inequality across panels. Four panels per treatment were immersed at 15 m depth in June 2014 and retrieved in March 2015, during which time settlement and growth of settlers was monitored[9]. Treatments were; (a) controls (no warming), (b) +1 °C, (c) +2 °C and (d) +1 °C in summer (Sep–Mar) only. Finally, we examined colonists of four controls immersed earlier (for a similar immersion duration) to enable comparisons of temporal variability.

**Photography and microscopy**. Upon retrieval, panels were photographed using a 1:1 macro lens and the images stitched together digitally. Species were identified by enlarging photographic images on monitors and comparing with relevant primary taxonomic literature. When greater resolution was needed, panels were directly examined using stereo microscopy. Each individual/colony was labelled using colour fill to identify the species (Figs. 2A, 2B). When this method had been repeated for each panel, the number of colonies for each species were recorded for each panel.

**Analysis of competition**. Competitive encounters were counted by eye, when >5% of the boundary of each colony was determined to be in contact another colony. Each encounter was defined as intraspecific if the two competitors were of the same identity and interspecific if the competitors comprised different species. These were quantified as the number of each competition type per area of each panel. The

measures of spatial competition made were the probability, density, complexity, the proportion of encounters, which were intraspecific, and the proportion of encounters involving the most frequent species (F. rugula). Our definitions of competition were, for each panel; (1) probability=number of competitive encounters/number of colonies; (2) density=number of competitive encounters cm$^{-2}$; (3) complexity=total number of different pairwise encounters types (e.g., competitor a meets b is one type of pairwise encounter); (4) Intraspecific=encounters where both competitor were of the same species/total number of competitive encounters; and the final metric was (5) those competitive encounters involving F. rugula/total number of encounters. A detailed close up of an example area of competition is shown in Fig. 4, with locations of spatial encounters marked numerically as counted.

**Statistics and reproducibility**. Our competitive encounter data (probability, density and complexity) was highly heteroskedatic, so Welch's one-way test with Games–Howell post hoc were used when assumptions of the ANOVA or Kruskal–Wallis tests were not met[19]. Statistical tests were completed using R software[20,21], specifically the *aov* and *oneway.test* function in the native *stats* package[20] and *posthocTGH* function in the *userfriendlyscience* package[22]. Similarity of species involved in competitive encounters and those involved in competitive encounters with F. rugula were analysed using ANOSIM and visualised using nMDS (functions *anosim* and *metaMDS* in the *vegan* package[22–24]). Significant global ANOSIM results were followed by pairwise tests and SIMPER analysis completed in Primer[25].

**Reporting summary**. Further information on research design is available in the Nature Research Reporting Summary linked to this article.

## Data availability
The original raw data are available online at https://github.com/Gaton1/Communications-Biology-Barnes-2020.

## Coda availability
The R codes used in data analysis are available online at https://github.com/Gaton1/Communications-Biology-Barnes-2020

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

## Acknowledgements

Thanks to Mark Preston for development of the settlement panels and technical advice, and to the Rothera Research Station assemblage for project support. This work was funded by NERC standard grant no. NE/J007501/1.

## Author contributions

D.K.A.B., G.V.A., S.A.M. and L.S.P. conceived the study, discussed the results, wrote the text and made the figures. G.V.A. analysed the data. D.K.A.B. led the team collecting the original data.

## Competing interests

The authors declare no competing interests.
