## [Peer Review File · Communications Biology]

Reviewers' comments:

Reviewer #1 (Remarks to the Author):

This manuscript investigates competition among newly recruited encrusting cyrptofauna on settlement plates in the shallows of the southern ocean under different temperature (warming) regimes. It is a new evaluation of data collected and published in 2017 for the investigation of structure and growth of these communities (citation #6). The methodology of the in situ deployment of warmed settlement plates to mimic predicted water temperature increases (although previously published) is innovative and interesting to the broader scientific community and the use of these data to analyze spatial competition among the cyrptofauna community is new.

The authors should expand upon the methods, specifically the plate design (this may consist of more clear/direct citing of the #6 paper, and their detailed methods supplement). The plate surface material is not specified (that I noticed), and the plate size differs slightly from the previous publication (10cm vs 9.8cm) – please correct. It is mentioned that the +1deg and +2deg were thoroughly tested pre-deployment, and that the desired temp increase was measured and consistent in the water immediately above the plate (2mm). There is no mention of the temperature of the plate itself, which I'd imagine would be slightly greater than the adjacent water temperature, but perhaps negligible (just curious more so than concerned). Finally, the statistical methods are not well defined. Please include the statistical package(s) used (I assume R and Primer) and what post hoc test(s) were used – perhaps Ganes-Howell – along with citations for the method/package developer(s). The use of Welch's test is warranted and appropriate for these data. Additionally, specifics should be included on how competition density and complexity were calculated. If text length is of concern, perhaps inclusion of a supplemental methods document with this level of detail is needed.

The reproducibility of the actual field work is complex, due to the extreme nature of the study area and the infrastructure requirements of the warming plates and power source. I have great respect for the complexity and logistical complications of working in this environment. The study includes just shy of 1 year of data, however the authors added the comparison to previous "control" no warming settlement data they had access to from a previous year which suggested that base settlement conditions were similar between at least two years of record. The author's make clear that this study accounts for competition observed within the first year of settlement, and that competition for a more established community may be altogether different. Additional years of data or replicates would certainly provide a more robust analysis (especially for the highly variable +2deg panels), but the data collected are sufficient for the results presented by the authors. Investigation of the temporal progression of competition intensity of a newly recruiting assemblage under the temperature treatments would be interesting, and it would appear that those data are available (per the ~ monthly photos from #6 paper). Especially since the authors mention the potential implications of increased growth for pioneer cryptofauna in the face of increasing scouring rates. i.e. at what point (with the increased growth rates of warming treatments) do you see the competition on treatment panels significantly greater than controls? Particularly because the sub-sampled "summer" treatment was (I assume) from the end of the panel deployment, i.e. most mature months of the community development from the study (making the inclusion of this comparison a bit convoluted-this should be made more clear). Again, this is my assumption because deployment and end dates/months were not provided (which I suggest adding)... I had to dig into the methods supplement from #6.

Figure 1, example of the control and +1deg panels is a helpful visual, while the inclusion of the image analysis highlighting the less abundant species is interesting, a visualization of the actual competition quantification would be more helpful for the reader. Both Fig 1 and 2 nMDS ordination plots show nice visual separation of treatment vs control along the #1 ordination (particularly for the Fig 2 community composition). It would be interesting to investigate/mention what the primary species are that are contributing to this ordination 1 variation and what potential functional impact beyond competition those species may have.

While this may be general knowledge in your field, the broad audience of Communications Biology

could benefit from citations for the statement "Bryozoans and other encrusting cryptofauna have proved strong model taxa for investigating spatial competition and likewise artificial substrata in the form of settlement panels, good experimental surfaces to investigate such encounter dynamics" (Lines 47-50).

I suggest including a final summary/implications statement at the end of the abstract wrapping up the findings. Something pointing out the potential importance of considering spatial competition and complexity to detect more nuanced and short-term impacts of localized climate change.

There are grammar and punctuation edits needed throughout the document. Recommend thorough editorial review of manuscript.

Line 57 – Define first use of acronym WAP.

Lines 151 and 152 – typical to cite figures in discussion?

The results presented in this manuscript are of interest and immediate relevance for a broad audience. While the study design and novel methods have been previously published, the focus on the impact of community structure and growth as a result of warming treatments on competition is a new analysis lens for these data. The better understanding of how competition (even if just among a young cryptofauna community) can change in the face of predicted increases in ocean warming is one of many pertinent next steps to better recognize, model, and potentially mitigate the imminent changes the oceans are likely to experience in the coming century as a result of global warming.

In summary, the paper is very focused and of relatively simplistic design which lends itself to clear and distinct analysis and conclusions. I found myself wanting more expanded discussion, and can see opportunity for additional research questions/analysis with data that I assume the researchers have access to. With that said, I can appreciate the succinct presentation of a novel research question which provides results that advance our understanding of the nuances of climate change/warming ocean impacts to cryptofauna of the southern ocean. I suggest more specifics be provided in the methods, and consideration to include a visualization of the competition quantification in Figure 1 if possible.

Reviewer #2 (Remarks to the Author):

1. Lines 14-15 in the abstract need to be re-written for clarity - ...might be detectable at an earlier stage when environments change more than individual species performance measures?
2. Line 16 missing "temperature"? - ...at ambient or warmed temperature to predict...
3. Line 25 change punctuation: multiple, complex, interacting physical changes
4. Line 26 "has included" or just "includes"
5. Lines 29-30 edit punctuation to: ..."individual species in isolation, but cumulative responses at assemblage and community levels, though poorly studied, will likely have greater consequences."
6. Provide some citations for information you present on the impacts of warming to biota and/or the few studies that address direct thermal effects.
7. Line 45 change "photographical stills" to "still photographs" and "Remotely" in ROV not "remote"
8. Lines 47-50 provide citation for work that has proved bryozoans as model organisms for this type of work
9. Line 54 missing "°C"
10. Line 57 define WAP before using the acronym
11. Line 60 use full genus and species names before abbreviating to *F. rugula*.
12. Line 79 edit for readability: ..."to predicted mid-century levels (+1°C)1..."
13. Clarify lines 96-99. Do you mean to say that there was a higher degree of intraspecific competition with warming but that the species involved did not change?
14. Line 104 missing comma "In contrast, species.."
15. Line 111-113 edit for readability: "...is proving difficult; the physical changes are complex in time and space, non-linear and non-conforming to model projections.9"
16. Line 113-114 is difficult to read I suggest rewording for readability, e.g. "Warming is just one

physical change, though an important one, that has already shown considerable biological and physical impacts to the Antarctic environment."

17. Line 116 consider using "individual and assemblage" to align with previous language in the introduction/results.

18. Lines 122-124 consider a forward statement defending your model and results rather than posing a rhetorical question. The manuscript would benefit from a strong, confident conclusion.

19. Line 134 remove "?" you are stating that there is a wider question regarding the changes observed not asking the question.

20. Lines 141 and 143 remove ")" at end of sentences.

21. Line 159 replace "situation" with "locations"

22. Line 167 is (9) supposed to be a reference? i.e. statement9

23. Line 169 italicize *in situ*

24. Figure 1: do the colors in right images for a and b mean something? Consider adding to legend if so.

25. Figure 2: it seems like panel A is just added as an afterthought. I recognize that it does not fit well within Figure 1, however, the study location would be better presented before other results if possible.

How was competition determined photographically? There are no image analysis methods provided! It is unclear from the methods how the data were generated and analyzed. For example, the definition of "competitive encounter" describes a "5% boundary in contact" and from Figure 1 it appears that the area of each individual was measured to ensure that their colony edges did not contact another but this is not provided in the methods at all. Further, the data "counts" were analyzed but counts of what? What program was used to analyze images? How were images collected? If these details have been provided in another manuscript, please cite appropriately. The results and conclusions should be understandable without reading detailed methods based on the format of this journal. This manuscript does not seem like it was written with this journal in mind and, therefore, suffers from a lack of detail and readability throughout. I suggest editing the introduction, results, and conclusions to be "stand alone". The manuscript also suffers from poor readability. There are many places (indicated in line items above) where the wording is confusing or unclear and could be more succinct. Significant grammatical errors also need to be addressed before acceptance of this manuscript.

Reviewer #3 (Remarks to the Author):

In this study, Barnes et al investigate patterns in competition of incrusting fauna in response to different warming regimes at the West Antarctic Peninsula. Marine benthic studies on patterns in competition in addition to standard community analysis in response to future climate scenarios are very scarce (if existing at all) and definitely new for the Southern Ocean. The competition analysis as opposed to general community analysis is very interesting and will influence thinking in the field. This study therefore surely merits the attention of the community. The results are presented in a clear fashion and are discussed against the relevant framework of climate warming. Based on the available information in the results, the conclusions are sound. My main comments are that (1) the argumentation for hypothesis 2 on less competition in response to 2°C warming as opposed to 1°C warming is not clear and (2) the methodology is largely unclear. With the current level of detail, it is not possible for other scientists to replicate the study. If these aspects and the comments specified below are taken into account, I believe this will be a very nice study for readers of *Communications Biology*.

Comments:

Abstract:

1. Please always define which pattern of competition is considered. E.g. l18-19: 'variance in competition': frequency, density or complexity of competition? Also check throughout ms if the considered competition pattern is specified.

2. L20: last sentence on temporal contrasts is not well connected to rest of abstract: introduce the time treatment earlier in abstract?

Introduction

3. L39: what exactly is meant with 'interactions in contest' when inter-and intraspecific competition is considered? Perhaps the term contest competition should be better defined.
4. L61: weak spatial competitor: why is *F. rugula* a weak spatial competitor and in which conditions?
5. L66: not clear which arguments led the authors to hypothesise that a 2°C warming would not lead to competition. Please elaborate.
6. L69: not clear how the studied patterns in competition would also describe the 'similarity among species involved in competitive pairings'. After reading results, I understand you refer to an analysis of similarity, but at first reading in the introduction I suggest to elaborate on this.

Results

7. L74: 5360 encrusting cryptofauna: not clear what this number refers to: colony density? Please state explicitly. Also, what was the species richness found on the panels?
8. L87: no supplementary figures were supplied in the review pane. Or do you refer to regular Figure 2E, F?
9. L99 ff: all these analyses are not described in methods.
10. L99: Similarity analysis ... where can we see these results? Please refer to figure or table.
11. L101: I don't find the difference between Figure 1F and Figure 2B so distinct. Could you give the figures the same dimensions?
12. L103-104: Here, only the overall ANOSIM results are presented. What are the results of the pairwise ANOSIMs? Did the 1C and 2C treatments differ from controls?

Discussion

13. L124: would competition in warming polar areas then become as 'severe' as in temperate areas? Can you, based on the results, speculate on possible shift from intraspecific to more interspecific competition as presented in Barnes & Neutel 2016?
14. L131: longer term effect: how would the communities evolve over time with sustained warming? Is it possible that interspecific competition decreases again once *F. rugula* starts to dominate? Can you relate this to the decreased intraspecific competition at 2°C warming (Figure 2E)?
15. L132: increase in what? Please specify which type of competition.
16. L141 ff: ice scouring is considered a covariate. Do the panels show traces of ice scouring? If so, would it be possible to include ice scouring as a covariate in the analysis to follow up on interacting warming/ice scouring effects on competition?
17. L154: how relevant are findings of this study on sessile organisms for other sessile and also mobile species?

Methods:

18. what are the dimensions of the used panels?
19. How were panels deployed, in which conditions, on what substrate?
20. L171: consistent temperature treatments: no temporal variability in the panels?
21. L175: how was temperature measured so closely to the panels?
22. L177 + L180: over which time period?
23. How were encrusting cryptofauna identified and quantified?
24. Figures mention image analysis. Which software was used?
25. L186: 'thus minimizing any potential food inequality': please specify how convective lift would influence food inequality for encrusting fauna (what is their food source).
26. L 187 Please also mention months
27. L189: how were competitive encounters quantified? Did you differentiate between intra-and interspecific competition?
28. L192: competitive encounter data? Which parameters were tested?
29. L193: which software was used for statistics? If R, which packages?

Figures

30. Figure 1: colour coding of the species?
31. Figure 1: x-axis scaling not clear: 'previous' is not explained in caption. X-axis of panel C is slightly cut
32. Figure 2 C, D are not referred to in text.

Textual comments:

L13: polar-regions: omit hyphen

L14-15: square brackets can be omitted

L25: sentence is cut and continues after full stop

L26: sentence contains two consecutive verbs ('has includes')

L54: '>1 warming regime' -> several warming regimes

L57: WAP: write in full at first mentioning

L60: F. rugula: write genus name in full

L91: were-> was

L142: omit bracket before full stop

L167: (9) as superscript

L169: what is a shore-based mains supply? The 'mains' is confusing.

L114: 'that it is already clear has considerable physical and biological impact'. Please reformulate.

L138: disentangling impact complexity ... is complex. Please reformulate.

L141: omit bracket

Responses to reviewers comments on COMMSBIO-20-1893

Our responses to the reviewers comments are detailed below each point (in blue)

Reviewer #1 (Remarks to the Author):

1) The authors should expand upon the methods, specifically the plate design (this may consist of more clear/direct citing of the #6 paper, and their detailed methods supplement). The plate surface material is not specified (that I noticed), and the plate size differs slightly from the previous publication (10cm vs 9.8cm) – please correct. It is mentioned that the +1deg and +2deg were thoroughly tested pre-deployment, and that the desired temp increase was measured and consistent in the water immediately above the plate (2mm). There is no mention of the temperature of the plate itself, which I'd imagine would be slightly greater than the adjacent water temperature, but perhaps negligible (just curious more so than concerned). Finally, the statistical methods are not well defined. Please include the statistical package(s) used (I assume R and Primer) and what post hoc test(s) were used – perhaps Ganes-Howell – along with citations for the method/package developer(s). The use of Welch's test is warranted and appropriate for these data. Additionally, specifics should be included on how competition density and complexity were calculated.

We have expanded our methods accordingly to include these points, details and given more definition to the statistical methodology.

2) At what point (with the increased growth rates of warming treatments) do you see the competition on treatment panels significantly greater than controls? Particularly because the sub-sampled "summer" treatment was (I assume) from the end of the panel deployment, i.e. most mature months of the community development from the study (making the inclusion of this comparison a bit convoluted-this should be made more clear). Again, this is my assumption because deployment and end dates/months were not provided (which I suggest adding)... I had to dig into the methods supplement from #6.

An interesting question but not a topic for this paper, we just analysed the images at the end of the experiment. It would require a very considerable body of work and time for someone to analyse every image from every month. We now give month of deployment and retrieval.

3) Figure 1, example of the control and +1deg panels is a helpful visual, while the inclusion of the image analysis highlighting the less abundant species is interesting, a visualization of the actual competition quantification would be more helpful for the reader.

We now show this in a new figure (4).

4) Both Fig 1 and 2 nMDS ordination plots show nice visual separation of treatment vs control along the #1 ordination (particularly for the Fig 2 community composition). It would be interesting to investigate/mention what the primary species are that are contributing to this ordination 1 variation and what potential functional impact beyond competition those species may have.

A line has been added to the results regarding the influence of *F. rugula* in significant differences identified during the nMDS analysis. Including additional detailed species information is beyond the scope of this paper.

5) While this may be general knowledge in your field, the broad audience of Communications Biology could benefit from citations for the statement “Bryozoans and other encrusting cryptofauna have proved strong model taxa for investigating spatial competition and likewise artificial substrata in the form of settlement panels, good experimental surfaces to investigate such encounter dynamics” (Lines 47-50).

Citations are now provided for this.

6) I suggest including a final summary/implications statement at the end of the abstract wrapping up the findings. Something pointing out the potential importance of considering spatial competition and complexity to detect more nuanced and short-term impacts of localized climate change.

We have added this but the journal guidance states that the abstract can be no more than 150 words, which it was already – so we have had to delete other text to keep within this.

7) There are grammar and punctuation edits needed throughout the document. Recommend thorough editorial review of manuscript.

Line 57 – Define first use of acronym WAP.

Lines 151 and 152 – typical to cite figures in discussion?

We have been through the ms thoroughly, defined the first use of WAP and found no journal guidance against citation of figures in discussion.

8) I suggest more specifics be provided in the methods, and consideration to include a visualization of the competition quantification in Figure 1 if possible.

As point 3.

Reviewer #2 (Remarks to the Author):

1. Lines 14-15 in the abstract need to be re-written for clarity - ...might be detectable at an earlier stage when environments change more than individual species performance measures?

Rewritten.

2. Line 16 missing “temperature”? - ...at ambient or warmed temperature to predict...

We added ‘temperature’.

3. Line 25 change punctuation: multiple, complex, interacting physical changes

Changed.

4. Line 26 “has included” or just “includes”

We went with includes.

5. Lines 29-30 edit punctuation to: ..."individual species in isolation, but cumulative responses at assemblage and community levels, though poorly studied, will likely have greater consequences."

Edited.

6. Provide some citations for information you present on the impacts of warming to biota and/or the few studies that address direct thermal effects.

Citations added (4-7).

7. Line 45 change "photographical stills" to "still photographs" and "Remotely" in ROV not "remote"

Changed.

8. Lines 47-50 provide citation for work that has proved bryozoans as model organisms for this type of work

Citations added (9-13)

9. Line 54 missing "°C"

Added.

10. Line 57 define WAP before using the acronym

Now defined on first use.

11. Line 60 use full genus and species names before abbreviating to *F. rugula*.

Now in full on first use.

12. Line 79 edit for readability: ..."to predicted mid-century levels (+1°C)1..."

Edited.

13. Clarify lines 96-99. Do you mean to say that there was a higher degree of intraspecific competition with warming but that the species involved did not change?

Yes and we have edited the line to make this clearer.

14. Line 104 missing comma "In contrast, species.."

Added.

15. Line 111-113 edit for readability: "...is proving difficult; the physical changes are complex in time and space, non-linear and non-conforming to model projections.9"

Edited as suggested.

16. Line 113-114 is difficult to read I suggest rewording for readability, e.g. "Warming is just one physical change, though an important one, that has already shown considerable biological and physical impacts to the Antarctic environment."

Reworded as suggested.

17. Line 116 consider using "individual and assemblage" to align with previous language in the introduction/results.

Changed to suggested text.

18. Lines 122-124 consider a forward statement defending your model and results rather than posing a rhetorical question. The manuscript would benefit from a strong, confident conclusion.

Changed as suggested.

19. Line 134 remove "?" you are stating that there is a wider question regarding the changes observed not asking the question.

Removed.

20. Lines 141 and 143 remove ")" at end of sentences.

Removed.

21. Line 159 replace "situation" with "locations"

Replaced.

22. Line 167 is (9) supposed to be a reference? i.e. statement⁹

Now put in superscript (as references should be).

23. Line 169 italicize in situ

Italicised

24. Figure 1: do the colors in right images for a and b mean something? Consider adding to legend if so.

We now say that rare species are shown in different colours.

25. Figure 2: it seems like panel A is just added as an afterthought. I recognize that it does not fit well

within Figure 1, however, the study location would be better presented before other results if possible.

We have now separated this and presented as new Fig. 1.

26. How was competition determined photographically? There are no image analysis methods provided! It is unclear from the methods how the data were generated and analyzed. For example, the definition of “competitive encounter” describes a “5% boundary in contact” and from Figure 1 it appears that the area of each individual was measured to ensure that their colony edges did not contact another but this is not provided in the methods at all. Further, the data “counts” were analyzed but counts of what? What program was used to analyze images? How were images collected?

We have added to the methods to detail these.

27. This manuscript does not seem like it was written with this journal in mind and, therefore, suffers from a lack of detail and readability throughout. I suggest editing the introduction, results, and conclusions to be “stand alone”. The manuscript also suffers from poor readability. There are many places (indicated in line items above) where the wording is confusing or unclear and could be more succinct. Significant grammatical errors also need to be addressed before acceptance of this manuscript.

We have been through the ms to try to improve these.

Reviewer #3 (Remarks to the Author):

My main comments are that (1) the argumentation for hypothesis 2 on less competition in response to 2°C warming as opposed to 1°C warming is not clear and (2) the methodology is largely unclear. With the current level of detail, it is not possible for other scientists to replicate the study. If these aspects and the comments specified below are taken into account, I believe this will be a very nice study for readers of Communications Biology.

We have tried to be clearer on these and expanded our methods.

Comments:

Abstract:

1. Please always define which pattern of competition is considered. E.g. l18-19: ‘variance in competition’: frequency, density or complexity of competition? Also check throughout ms if the considered competition pattern is specified.

We have added competition type.

2. L20: last sentence on temporal contrasts is not well connected to rest of abstract: introduce the time treatment earlier in abstract?

We have restructured our abstract.

Introduction

3. L39: what exactly is meant with 'interactions in contest' when inter-and intraspecific competition is considered? Perhaps the term contest competition should be better defined.

We have rewritten this and defined contest competition.

4. L61: weak spatial competitor: why is *F. rugula* a weak spatial competitor and in which conditions?

We have defined why we describe it as a weak competitor.

5. L66: not clear which arguments led the authors to hypothesise that a 2°C warming would not lead to competition. Please elaborate.

We now elaborate on this to explain.

6. L69: not clear how the studied patterns in competition would also describe the 'similarity among species involved in competitive pairings'. After reading results, I understand you refer to an analysis of similarity, but at first reading in the introduction I suggest to elaborate on this.

We have elaborated on this (as per other referees comments).

Results

7. L74: 5360 incrusting cryptofauna: not clear what this number refers to: colony density? Please state explicitly. Also, what was the species richness found on the panels?

Yes we have added 'colony density' and report species richness '9-16 species'.

8. L87: no supplementary figures were supplied in the review pane. Or do you refer to regular Figure 2E, F?

Our error, yes we referred to former fig 2e,f – now corrected.

9. L99 ff: all these analyses are not described in methods.

These are now described (in methods).

10. L99: Similarity analysis ... where can we see these results? Please refer to figure or table.

Reference to figure added.

11. L101: I don't find the difference between Figure 1F and Figure 2B so distinct. Could you give the figures the same dimensions?

They are now placed adjacent to each other and similar in size.

12. L103-104: Here, only the overall ANOSIM results are presented. What are the results of the pairwise ANOSIMs? Did the 1C and 2C treatments differ from controls?

Pairwise ANOSIM results now given.

Discussion

13. L124: would competition in warming polar areas then become as 'severe' as in temperate areas? Can you, based on the results, speculate on possible shift from intraspecific to more interspecific competition as presented in Barnes & Neutel 2016?

We now speculate that, but it is hard to know.

14. a) L131: longer term effect: how would the communities evolve over time with sustained warming? b) Is it possible that interspecific competition decreases again once *F. rugula* starts to dominate? c) Can you relate this to the decreased intraspecific competition at 2°C warming (Figure 2E)?

a) We don't know, this would be a big guess. b) unlikely – *F. rugula* only dominates in the absence of other species, when they arrive they nearly all overgrow it. c) intraspecific competition at 2C is not 'decreased' it is insignificantly different (to either 1C or controls).

15. L132: increase in what? Please specify which type of competition.

We now state 'competition probability and density'.

16. L141 ff: ice scouring is considered a covariate. Do the panels show traces of ice scouring? If so, would it be possible to include ice scouring as a covariate in the analysis to follow up on interacting warming/ice scouring effects on competition?

We have added a line to say that meeting between icescour and panels = crushed panel!

17. L154: how relevant are findings of this study on sessile organisms for other sessile and also mobile species?

Hard to know without a heated marine mesocosm to investigate those other species..

Methods:

18. what are the dimensions of the used panels?

We now state this.

19. How were panels deployed, in which conditions, on what substrate?

We now state this.

20. L171: consistent temperature treatments: no temporal variability in the panels?

The temperature treatments stayed consistently +1 and +2 C above ambient. Ambient temperature varied continuously (though only by ~3C all year round).

21. L175: how was temperature measured so closely to the panels?

We have added a line in methods to state this.

22. L177 + L180: over which time period?

We have added the time period of the experiment.

23. How were encrusting cryptofauna identified and quantified?

We have added this detail – identified using primary literature and quantified by counting within each set panel area.

24. Figures mention image analysis. Which software was used?

The image analysis has been described in more detail in the methods.

25. L186: ‘thus minimizing any potential food inequality’: please specify how convective lift would influence food inequality for encrusting fauna (what is their food source).

We have added a line to state this.

26. L 187 Please also mention months

We now give months.

27. L189: how were competitive encounters quantified? Did you differentiate between intra-and interspecific competition?

As per point 23) we now explain more and yes we did differentiate between inter and intra competitive encounters.

28. L192: competitive encounter data? Which parameters were tested?

We have added parameters in parentheses.

29. L193: which software was used for statistics? If R, which packages?

Now given.

Figures

30. Figure 1: colour coding of the species?

As per referee 2 point we have added that this was to show rare species.

31. Figure 1: x-axis scaling not clear: ‘previous’ is not explained in caption. X-axis of panel C is slightly cut

Figure redone.

32. Figure 2 C, D are not referred to in text.

Figures now referred to in text.

Textual comments:

L13: polar-regions: omit hyphen

L14-15: square brackets can be omitted

L25: sentence is cut and continues after full stop

L26: sentence contains two consecutive verbs ('has includes')

L54: '>1 warming regime' -> several warming regimes

L57: WAP: write in full at first mentioning

L60: F. rugula: write genus name in full

L91: were-> was

L142: omit bracket before full stop

L167: (9) as superscript

L169: what is a shore-based mains supply? The 'mains' is confusing.

L114: 'that it is already clear has considerable physical and biological impact'. Please reformulate.

L138: disentangling impact complexity ... is complex. Please reformulate.

L141: omit bracket

All points corrected.

David Barnes on behalf of co-authors.

REVIEWERS' COMMENTS:

Reviewer #1 (Remarks to the Author):

Dear Authors,

The expanded methods sufficiently address my comments/concerns. Thank you for your revisions, i look forward to seeing the manuscript in print.

Reviewer #2 (Remarks to the Author):

This manuscript represents a unique, controlled study on the effects of warming on the competitive interactions among encrusting polar fauna. The authors conclude that warming predicted by mid-century causes increased competitive interactions among encrusting fauna and indirect changes to the complexity of these interactions. The study builds from the novel experimental design reported previously and expands the data collected in a new analysis. I think this fact highlights the utility of complex studies and I commend the authors for using their dataset to further explore intricate interactions. Though there is additional analysis that could be conducted (as with any study) the present conclusions are warranted for publication. The focus on competitive interaction changes with climate warming is unique and an important contribution to the field emphasizing the direct and indirect effects of warming on hard-bottom communities. The study is an example of how controlled experimentation can be a useful tool to understanding the nuances of faunal interactions that scale to community-level changes across the Antarctic and predict future community alterations. The findings reported here are integral to understanding changes to shallow-water communities in the Antarctic with respect to temperature and the methods presented are useful for application in aquatic environments globally.

The revised manuscript now offers sufficient detail of methods by which the authors enumerated competitive interactions and statistically tested their results that greatly improve reproducibility and robustness of the study. Specifically, the addition of figure 1 clarifies the study region earlier in the text which is necessary for providing the readers with regional context of the study, and figure 4 provides detailed visualization of competitive interaction identification. Inclusion of figure 4 and added details of the methodology reduces the subjective nature inherent in image analysis and greatly bolsters the findings of the study. The authors have done a nice job addressing all referee comments and suggested edits. Overall, the manuscript is more readable, thorough, and reflects the careful design of the study. I am pleased with the revised manuscript, commend the authors for a thorough revision, and recommend publication of this work.